# Ising meson spectroscopy on a noisy digital quantum simulator

Christopher Lamb[1] ✉, Yicheng Tang[1], Robert Davis[1] & Ananda Roy [1] ✉

Quantum simulation has the potential to be an indispensable technique for the investigation of non-perturbative phenomena in strongly-interacting quantum field theories (QFTs). In the modern quantum era, with Noisy Intermediate Scale Quantum (NISQ) simulators widely available and larger-scale quantum machines on the horizon, it is natural to ask: what non-perturbative QFT problems can be solved with the existing quantum hardware? We show that existing noisy quantum machines can be used to analyze the energy spectrum of several strongly-interacting 1+1D QFTs, which exhibit non-perturbative effects like 'quark confinement' and 'false vacuum decay'. We perform quench experiments on IBM's quantum simulators to compute the energy spectrum of 1+1D quantum Ising model with a longitudinal field. Our results demonstrate that digital quantum simulation in the NISQ era has the potential to be a viable alternative to numerical techniques such as density matrix renormalization group or the truncated conformal space methods for analyzing QFTs.

Investigation of non-perturbative phenomena in strongly interacting quantum field theories (QFTs) remains one of the outstanding challenges of modern physics. Despite impressive progress, ab-initio lattice computations of properties of arbitrary QFTs will likely remain beyond the computational capabilities of the most powerful classical computer. This is due to the exponentially-growing Hilbert space which poses an insurmountable challenge for exact simulation of these lattice models. Nevertheless, quasi-exact or approximate classical computing techniques have been successful in investigating a large family of quantum many-body systems that realize QFTs in the scaling limit. For example, states with 'relatively low entanglement' can be efficiently represented using tensor networks[1–4]. This has allowed high-precision investigation of low-energy states of one-dimensional, local, gapped and gapless Hamiltonians using matrix product states[5] as well as their two-dimensional generalizations[6–8].

Despite the success of classical computing methods, simulation of high-energy states or non-equilibrium dynamics remains challenging even for generic, strongly-interacting 1+1D quantum systems. This is due to the rapid growth of entanglement between different subsystems[9–12]. Quantum simulation, both analog and digital, have the potential to become a viable alternative to the aforementioned approaches for tackling these questions[13,14]. Analog quantum simulation involves tailoring a given quantum system to emulate a specific target model[15–20] and has been remarkably successful in analyzing a wide range of physical problems. However, it is gate-based, error-corrected, digital quantum simulation that is the long-term, universal solution for simulation of QFTs[21]. In this approach, lattice regularizations of QFTs can be encoded into registers of error-corrected qubits with suitable initial state preparation, time evolution and measurement protocols. To that end, protocols for digitization of scalar fields[22,23] and quantum algorithms for computation of scattering processes in scalar[24] and fermionic[25] QFTs have already been proposed.

However, implementing such a quantum simulation protocol on existing quantum hardware remains a daunting challenge. In spite of the recent progress in the fabrication and manipulation photonic and solid-state systems, the number of available qubits across the different quantum computing platforms remains around $10^2$ with modest coherence times. Given the large overhead in the number of required qubits to implement quantum error correcting codes[26], it is an open problem to implement error-corrected digital simulation of QFTs in the near term. This raises the question: what non-perturbative QFT problems can be tackled with current Noisy Intermediate Scale Quantum (NISQ) machines?

In this work, a superconducting NISQ simulator is used to compute the energy spectra of certain strongly-interacting QFTs that arise as scaling limits of one-dimensional quantum spin chains. The latter

[1]Department of Physics and Astronomy, Rutgers University, Piscataway, NJ, USA. ✉e-mail: cdl92@physics.rutgers.edu; ananda.roy@physics.rutgers.edu

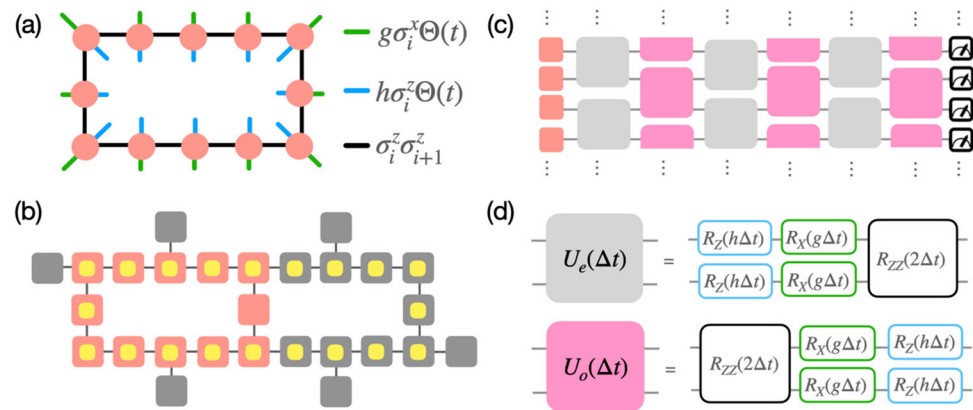

**Fig. 1 | Schematic of the Ising meson spectroscopy experiment. a** The coral circles indicate spin-1/2 sites. The black, green and blue bonds respectively correspond to the ferromagnetic interaction, transverse and longitudinal fields respectively. The quench protocol involves turning on, at $t = 0$, the longitudinal and transverse field strengths: $g \leq 1$, $h < 1$ in the Hamiltonian $H$ [Eq. (1)]. The system is initially prepared in the ground state of $H$ with $g = h = 0$: $|\uparrow, \ldots, \uparrow\rangle$. **b** Schematic of the qubit layout in IBM's `ibm_auckland` and `ibmq_mumbai` quantum simulators.

Of the available 27 qubits (in gray), a periodic chain of 12 qubits (in coral) and 20 qubits (in yellow) were chosen for the quench experiments. **c** Schematic of the trotterized unitary time-evolution generated by $H$. The action of the full unitary operator is decomposed into the unitary operators: $U_e$ and $U_o$. At the end of the time-evolution, single qubit measurements are performed in the $\sigma_y$ basis. **d** The decompositions of $U_e$ and $U_o$ in terms of single and two qubit gates are shown.

QFTs offer a valuable playground for investigation of a wide range of non-perturbative phenomena such as 'quark confinement'[27] and 'false vacuum decay'[28] that are typically associated with QFTs in higher-dimensions. Importantly, there are two practical advantages: i) these spin-chains can be directly mapped to arrays of qubits without additional overhead involved in discretizing the local lattice degree of freedom, ii) several properties of these spin-chains can be investigated with existing classical numerical techniques (tensor networks) and analytical methods (Bethe ansatz and bosonization). This enables benchmarking NISQ simulators before using them for investigation of problems that truly lie beyond the reach of the classical computers.

## Results

The goal of this work is to demonstrate the aforementioned using the paradigmatic example of the one-dimensional quantum Ising model in the presence of a longitudinal field[29]. The Hamiltonian is

$$H = -\sum_{j=1}^{L} \left( \sigma_j^z \sigma_{j+1}^z + g\sigma_j^x + h\sigma_j^z \right), \tag{1}$$

where $g$, $h$ are the strengths of the transverse and longitudinal fields and we have chosen periodic boundary conditions. We consider the case $g \leq 1$, $h < 1$. The presence of the longitudinal field introduces a confining potential between the domain-wall excitations of the Ising model. This leads to formation of 'mesonic' bound states, analogous to hadron-formation in quantum chromodynamics[30,31]. The energy spectrum has been analyzed using semi-classical[32,33], truncated conformal space[34–38] and exact diagonalization[39] approaches. While signatures of confinement have been observed for this model in a NISQ simulator[40] (see also ref. 41 for analysis of a related model), digital quantum simulation of the energy spectrum has remained elusive. This is performed in this work using a quench experiment on IBM's `ibm_auckland` simulator which is one of the IBM Quantum Falcon Processors. The quantum simulation protocol is first tested using the Qiskit software package[42] and benchmarked against exact computations. Subsequently, it is implemented on the noisy quantum hardware.

The quantum simulation protocol is as follows (see Fig. 1). First, the qubits are prepared in the state: $|\uparrow, \ldots, \uparrow\rangle$, which is one of the two degenerate ground state of the Hamiltonian $H$ with $g = h = 0$. Next, a global quench is performed to $0 < g \leq 1$, $h < 1$ on the 12 qubit loop (in coral in Fig. 1) of the `ibm_auckland` simulator. The quench is

implemented in the NISQ hardware by the application of single and two-qubit gates [Fig. 1c, d][43] that together give rise to the unitary operator $U = e^{-iH\Delta t}$. Here, $\Delta t$ is the size of the time-step. The rest energy spectrum of the mesonic excitations formed due to the confinement of fermionic Ising domain-walls manifests itself in the observables that involve only single-qubit operators[39]. This is particularly suitable for implementation in a NISQ machine where single qubit measurements are performed with relatively low error. In particular, in this work, the single-qubit observable $\langle \sigma_j^y(t) \rangle$, $j$ being the site-index, is measured during the course of this evolution. Note that the meson rest energy spectrum can be also computed by measuring $\langle \sigma_j^x(t) \rangle$ or $\langle \sigma_j^z(t) \rangle$. On a noisy simulator, the best noise-resilience for the energy computation was obtained for $\langle \sigma_j^y(t) \rangle$. The same choice was found to be optimal with regards to the errors arising from trotterization of the continuous time-evolution.

Figure 2 shows the results for the quenches from $g = h = 0$ to $g \leq 1$, $h = 0.3$. The top panels show the measured variation of $\sigma^y(t) = \sum_j \langle \sigma_j^y(t) \rangle / L$ with time. The solid green curves act as reference and are obtained from exact computation of the time-evolution of the chain of qubits with $\Delta t = 0.01$. The dashed orange curves are obtained from numerical simulations on a noiseless Qiskit simulator for $\Delta t = 0.4$. In this case, the expectation value $\langle \sigma_j^y(t) \rangle$ was computed for each qubit at each time-step by measuring it in the corresponding basis and averaging over 8192 shots. The close agreement of the noiseless simulation results for the two choices of $\Delta t$ indicates a small trotterization error for this observable. The quantum simulation experiment on `ibm_auckland` was performed with the same parameters as the noiseless simulations (experimental data as blue diamonds in Fig. 2). To mitigate the effects of qubit and gate errors on the quantum hardware, a pulse-efficient quantum circuit was implemented (see Supplementary Note 1) in addition to the in-built Qiskit twirled readout error extinction and dynamical decoupling schemes. The bottom panels show the absolute values of the corresponding Fourier transforms. The rest energies of the mesons are inferred from the location of the peaks, with the dashed vertical lines being results obtained using exact diagonalization in the zero momentum sector. Note that the quality of the data from the quantum hardware for $g = 1$ is worse than the $g < 1$ cases. This is due to the lower gap between the ground and first excited states of the energy spectrum. This makes the experiments more susceptible to the noise arising due to trotterization and noise in the `ibm_auckland` simulator. In addition, for the $g = 1$ case, the numerical values of $e_{13}$ and $e_1$ are close to each other (see dashed

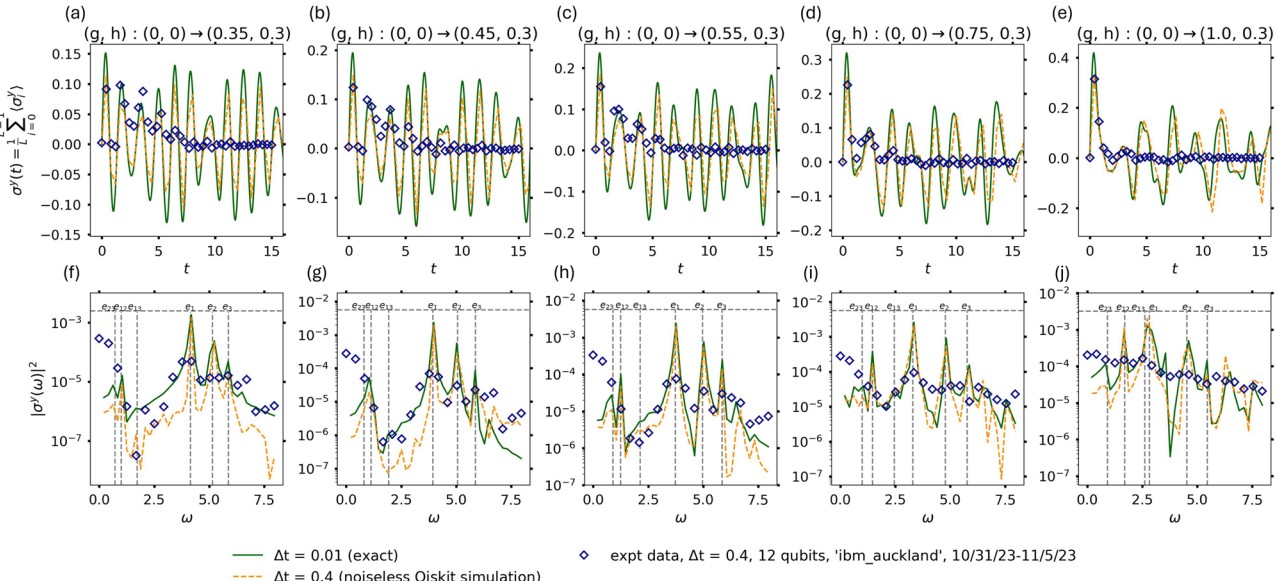

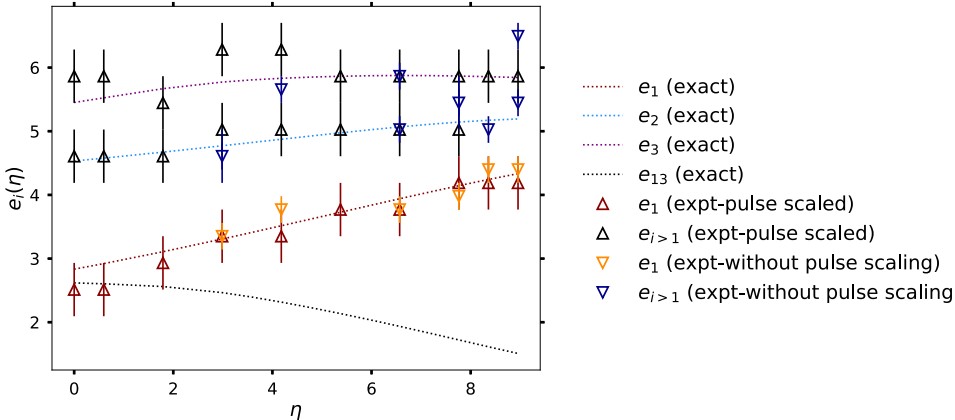

**Fig. 2 | Ising meson spectroscopy simulation and experimental results for quench from $g = h = 0$ using pulse-efficient quantum circuits.** The Qiskit twirled readout error extinction and dynamical decoupling schemes were used to mitigate the effects of noise present in the `ibm_auckland` simulator. **a**–**e** Results for $\sigma^y(t) = \sum_j \langle \sigma_j^y(t) \rangle / L$ for different quench parameters are shown. A periodic chain with length $L = 12$ qubits (in coral in Fig. 1) was used for quenches to $g \leq 1$ with $h = 0.3$. The results obtained using exact simulation of the time evolution of $L$ qubits with $\Delta t = 0.01$ are shown with green solid lines. The noiseless numerical Qiskit simulation results are performed for $\Delta t = 0.4$ (orange dashed line). The expectation values for the noiseless simulations are computed by trotterized time-evolution and making measurements in the $\sigma^y$ basis for each qubit after every time-step and averaging over 5 runs with 8192 shots each. The experimental data from the quantum hardware are shown as blue diamonds. **f**–**j** The squared absolute values of the Fourier transforms $\sigma^y(\omega)$ are shown. The peaks in the latter correspond to the rest energies ($e_n$), of the mesonic excitations labeled by $n$, and their differences ($e_{mn}$). The associated error bars are obtained from the uncertainty in the peak location. The gray dashed lines correspond to the exact diagonalization results obtained in the zero momentum sector. As shown in the bottom panels, the lowest meson energy is clearly discernible except for the rightmost panel, where the close proximity of the numerical values of $e_1$ and $e_{13}$ prevent an unambiguous identification.

**Fig. 3 | Variation of the meson energies obtained from quantum simulation experiments performed on the `ibm_auckland` (with pulse-scaling) and `ibmq_mumbai` (without pulse-scaling) simulators.** The results (triangles) are shown for a quench from $g = h = 0$ to $g \in [0.25, 1.0]$ and $h = 0.3$. The corresponding predictions from exact diagonalization of the 12-site qubit chain keeping only the translation-invariant states are shown as dotted lines. The meson energies are shown as a function of the dimensionless parameter $\eta = 2\pi(1 - g)/h^{8/15}$. The peaks for the different meson energies were identified by computing the $\sigma^y(\omega)$ for quenching to different choices of $g$ (Fig. 2). Note that the obtained quantum data did not always permit unambiguous identification of the second and third meson energies. Furthermore, for the smallest three values of $\eta$, the difference between the first and the third meson energies (black dotted line) is also close to the obtained peak in the spectral function (see also rightmost panels of Fig. 2). Smaller trotterization steps would enable identification of the lowest meson energies for these choice of parameters, but additional error-mitigation techniques will be necessary.

lines in bottom right panel of Fig. 2). This prevents an unambiguous identification of the lowest meson energy in this particular case already with the exact computation performed using $\Delta t = 0.01$ (solid green lines). We note that even though the obtained amplitude of the time-series data decays rapidly due to decoherence and gate errors, the frequency, which contains the information of the meson energies, is still discernible. This is true also for other analyzed models, see Supplementary Note 1.

Figure 3 shows the variation of the meson energies obtained from different quench experiments performed on the `ibm_auckland` simulator. In every experiment, the system was initialized in the ground state of $g = h = 0$. Subsequently, a quench was performed to $h = 0.3$ and $g$ varying between 0.25 and 1.0. The obtained meson energies are shown as diamonds. The corresponding predictions from the exact diagonalization of the qubit chain in the zero momentum sector are shown as dotted lines. Notice that the noisy data did not

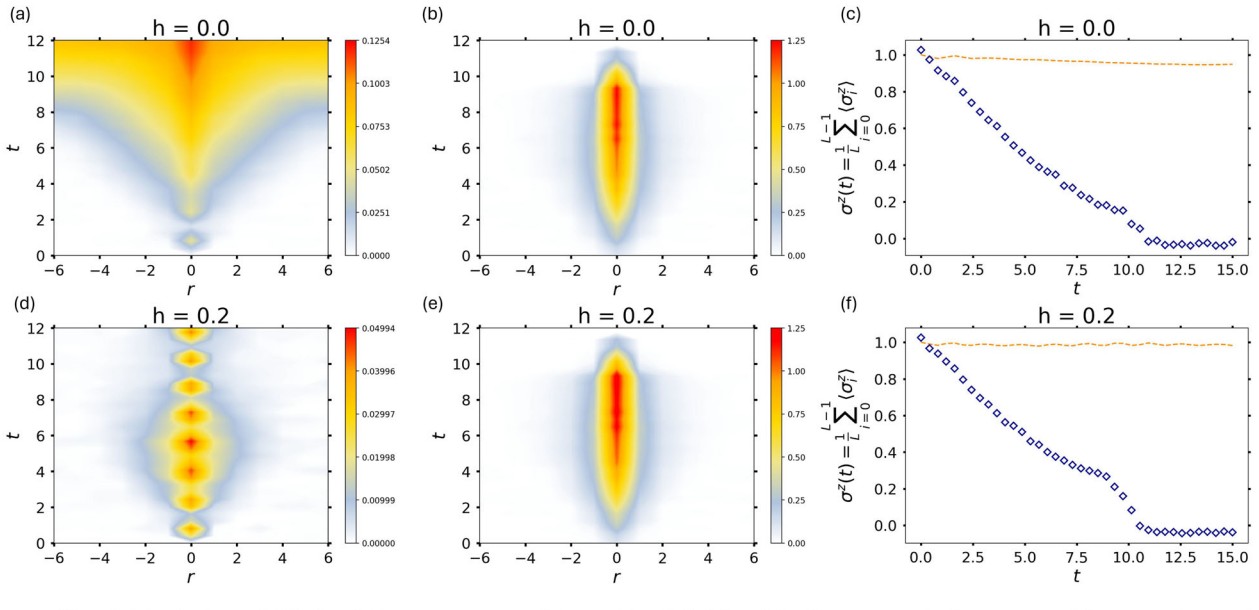

----- Δt = 0.4 (noiseless qiskit simulation)    ◇    expt data, Δt = 0.4, 12 qubits, 'ibmq_mumbai', average of 5 reps, pulse-scaled

**Fig. 4 | Results for $G_z(r, t)$ and the $\langle \sigma^z(t) \rangle$ for a quench to $g = 0.25$, $h = 0$ and $g = 0.25$, $h = 0.2$.** The noiseless simulation results for $G_z(r, t)$ are shown in panels **a**, **d**. The corresponding results for $\sigma^z(t)$ are shown with orange dashed lines in panels **c**, **f**. The simulation parameters are the same as in Fig. 2. The results of the quantum simulation experiments on the `ibmq_mumbai` simulator are shown in the panels **b**, **e** and with blue diamonds in panels **c**, **f**. With the current noisy gates and qubits on the quantum hardware, a distinct signature of confinement could not be discerned. This is consistent with the rapid decay of $\langle \sigma^z(t) \rangle$ (blue diamonds in the right panels). Additional error mitigation techniques and improved quantum hardware can be used to obtain more reliable signatures of the confinement in the correlation functions.

always permit unique identification of the second and third meson energies. Furthermore, for the three smallest values of $\eta$, the lowest meson energy ($e_1$) is close in numerical value to the difference between the first and third meson energies ($e_{13}$). To uniquely identify between $e_1$ and $e_{13}$ for these three values of parameters, smaller trotterization steps and additional error mitigation techniques will be necessary.

Figure 4 shows the results for the connected correlation function $G_z(r, t) = \langle \sigma_i^z(t) \sigma_{i+r}^z(t) \rangle - \langle \sigma_i^z(t) \rangle \langle \sigma_{i+r}^z(t) \rangle$. The latter contains signatures of the confinement of the domain-walls into mesonic excitations[39]. The results of the noiseless Qiskit simulations are shown in the left panels. The simulation parameters are kept as in Fig. 2. For $h = 0$, the correlation function exhibits a characteristic broadening due to the propagation of the energy-carrying domain wall excitations[9] until the separation between the latter is of the order of the system size (top left panel). For $h \neq 0$, as the domain-walls get separated, they feel the confining force that leads to the formation of the mesonic bound states. This results in oscillatory behavior of the correlation function[39] (bottom left panel). The corresponding expectation value $\sigma^z(t) = \sum_j \langle \sigma_j^z(t) \rangle / L$ for the noiseless simulations are shown with the dashed orange line in the right panels. The data from the quantum hardware is shown in the middle panels. With the current noise levels in the gates and modest lifetimes of the qubits of the `ibmq_mumbai` simulator, signatures of confinement could not be reliably discerned for the noisy experimental data for the correlation function. This is consistent with the measured rapid decay of $\sigma^z(t)$ shown with blue diamonds in the right panel (compare the oscillations obtained for $\langle \sigma^y(t) \rangle$ shown in Fig. 2). Additional error mitigation techniques and improved quantum hardware could be used to obtain signatures of confinement in the correlation functions.

## Discussion

In summary, this work highlights the potential of NISQ simulators for investigation of non-perturbative problems in strongly-interacting QFTs. In particular, a noisy IBM simulator is used to compute the energy-spectrum of the mesonic excitations occurring in the one-dimensional Ising model with a longitudinal field. The demonstrated quantum simulation scheme can be straightforwardly generalized to analyze a wide family of 1+1D QFTs. A natural extension is the quantum sine-Gordon model perturbed by a cosine potential with twice the periodicity. This model can be realized as the scaling limit of the XYZ spin chain[44] in the presence of a longitudinal field. In the latter model, the free Ising domain walls are replaced by interacting sine-Gordon solitons[45-48]. For details of the simulation protocol and corresponding results, see Supplementary Note 1. With improved noise properties, larger sizes, and more sophisticated error-mitigation techniques that are likely to be available in the coming years, it is conceivable that a wider range of QFTs will become amenable to quantum simulation on quantum hardware. This can lead to investigation of a wide range of exotic QFT phenomena ranging from false-vacuum decays[49,50] to Bloch oscillations[51,52] in table-top quantum simulation experiments.

## Methods

The experiment is set up by initializing the $|0,...,0\rangle$ ground state, and a unitary comprised of the trotterized time evolution of the Hamiltonian is applied for each $\Delta t$. After each Trotter step, the desired observable is measured.

We implement the quench protocol by a Trotter decomposition of the time evolution for the Hamiltonian of Eq. (1)

$$
\begin{aligned}
e^{-iHt} &= e^{-i(H_{zz} + H_x + H_z)t} \\
&\approx \prod_1^n e^{-iH_{zz}\delta t} e^{-iH_x\delta t} e^{-iH_z\delta t}
\end{aligned}
\tag{2}
$$

where $H_{zz} = -J\sum_j Z_j Z_{j+1}$, $H_x = -g\sum_j X_j$, and $H_z = -h\sum_j Z_j$. The unitary was decomposed further by changing the single qubit gates to the second-order Trotter decomposition while leaving the two-qubit rotation gates as first order.

$$
e^{-iHt} \approx \prod_1^n e^{-iH_z\frac{\delta t}{2}} e^{-iH_x\frac{\delta t}{2}} e^{-iH_{zz}\delta t} e^{-iH_x\frac{\delta t}{2}} e^{-iH_z\frac{\delta t}{2}}
\tag{3}
$$

$$e^{-iH_{zz}\delta t} \rightarrow R_{zz}(-2J\delta t), e^{-iH_x\delta t} \rightarrow R_x(-g\delta t), \qquad (4)$$

$$e^{-iH_z\delta t} \rightarrow R_z(-h\delta t) \qquad (5)$$

The two-qubit rotation gate, $R_{zz}$, was applied to the even and odd qubit indices to run as many operations in parallel as possible. Running the gates in parallel shortened the overall depth of the circuit to execute more gates during the qubit lifetimes.

Once the unitary gate was constructed, the $R_{zz}$ was scaled into a more pulse-efficient gate[53,54] to execute on IBM Quantum's machines. IBM Quantum uses a calibrated *CNOT* gate built from a $R_{zx}(\pi/2)$ rotation, each with about $\approx 1\%$ error, and is implemented by an echoed cross-resonance (CR) pulse. We reduce the rotation angle by changing the $R_{zz}$ gate to the pulse-efficient equivalent, shown in Supplementary Fig. 1, which scales the area of the CR pulse, effectively reducing the compounding errors. The same set of experiments were also performed without pulse-scaling, see Supplementary Note 1 for details.

To run the circuits on IBM's 27-qubit Falcon processor, `ibmq_mumbai` and `ibm_auckland`, the circuit needed to be transpiled for the qubits to map to the qubit loops available on the processor. This was done by passing in the initial layout of numbered qubits corresponding to available chains from the connectivity graphs. When the circuits were being transpiled, we added dynamical decoupling[55] by setting the optimization level to 3. Dynamical decoupling is a class of error mitigation techniques that add spin echoes to the transpiled circuits. The spin echoes send pulses to idle qubits, returning the qubits to their original states and undoing potentially adverse effects from nearby qubits.

The circuits were executed using the qiskit runtime primitive, Sampler. Using Sampler, we applied matrix-free measurement mitigation (M3)[56] to our circuits by setting the resilience level to 1 in the run options.

## Data availability

The time-series and correlation function data used in this study are available in the GitHub database under https://github.com/physicslamb/Ising_Meson_Spectroscopy_on_a_Noisy_Digital_Quantum_Simulator_Data.

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

## Acknowledgements

We acknowledge discussions with Nicholas Bronn, Robert Konik, and Olivia Lanes and the use of IBM Quantum services for this work. The views expressed are those of the authors, and do not reflect the official policy or position of IBM or the IBM Quantum team. This work was supported by the U.S. Department of Energy, Office of Basic Energy Sciences, under Contract No. DE-SC0012704.

## Author contributions

C.L. performed the quench experiments on IBM machines and wrote a portion of the manuscript. Y.T. and R.D. performed the computations necessary for performing the quantum simulation experiments. A.R. conceived and supervised the project and wrote a portion of the manuscript.

## Competing interests

The authors declare no competing interests.
