## [Peer Review File · Nature Communications]

REVIEWER COMMENTS

Reviewer #1 (Remarks to the Author):

The paper carries out a digital quantum simulation of basic quantum field theory models mapped onto an Ising spin model with transverse field.

In my view, the most noteworthy aspect of the paper is the complexity of the experiments carried out, with 12 and 20 qubits, in this context of quantum simulation of QFTs. Naturally, this is based on employing the available IBM resources, and not so much on experimental technology of the coauthors' group. Moreover, quantum simulating an Ising model is not such a novel achievement. Several experiments have been already carried out, both in trapped ions and superconducting circuits, of Ising-like models with digital quantum simulators, such as, e.g., experiments in the Innsbruck trapped ion lab, as well as in ETH and Google superconducting circuit labs.

According to my previous comments, in my view this paper would be more suitable for an archival journal.

Reviewer #2 (Remarks to the Author):

This work considers the potential of NISQ devices as simulators of non-perturbative quantum field theories. While problems directly relevant to high energy physics, such as QCD, are way beyond the capabilities of present quantum computers, analogous problems highly relevant to condensed matter physics can be studied in 1+1 dimensional systems. The authors choose a prominent example of these systems, namely, the one-dimensional quantum Ising model, known to have confinement analogous to QCD in its ferromagnetic phase. The resulting meson spectrum can be extracted from quench spectroscopy. The quantum simulation is tested and benchmarked against the theoretical results using the Qiskit package, then implemented on the NISQ hardware.

The problem addressed by the authors is timely and important; the results are interesting and substantially advance our understanding. The paper is well-written, the exposition is clear, and the authors used state-of-the-art methods to investigate the problem.

In my opinion, before eventual acceptance, the authors should consider the following issues and implement the necessary changes to address them:

1. The authors claim that in the NISQ system, the mass of the first meson can always be unambiguously obtained. However, Fig. 2 in itself does not clearly support this statement. Supposing that one only has the experimental data, it needs to be demonstrated that standard peak finding and fitting algorithms do indeed locate the correct position of the first meson peak and also do the same for other data depicted in Fig. 3. An additional question is whether a peak finder algorithm would also produce false positives.

I recommend that the authors perform such an analysis and add the relevant results, as this would make the support of their main statements much more robust.

2. I also find some crucial omissions in the literature considered in the paper; their inclusion is also relevant to emphasize the scope of the present work and future avenues of exploration.

The first application of quench spectroscopy to this model appeared in the paper by Rakovszky et al. in *Nuclear Physics B* 911 (2016) 805-845, in which both the spin chain problem and the corresponding QFT was treated, using iTEBD for the spin chain and TCSA for the QFT.

An additional phenomenon of interest is Bloch oscillations and Stark localisation, which was addressed by A. Leroze et al. in *Phys. Rev. B* 102, 041118(R) (2020), and by O. Pomponio et al. in *SciPost Phys.* 12, 061 (2022). In particular, the latter work performs quench spectroscopy to reveal the characteristic frequency signature of Bloch oscillations. This shows that this phenomenon can also be studied with the methods of the present work, and it would certainly be very promising to consider it for follow-up investigations.

The authors also mention the false vacuum decay and an interesting physical effect in this system. This was considered on the spin chain in G. Lagnese et al., *Phys. Rev. B* 104, L201106 (2021) and in the QFT in M. Lencses et al., *Phys. Rev. D* 106, 105003 (2022). This leads to another interesting question: whether the NISQ hardware used in this work can address the false vacuum decay.

In conclusion, I think the results warrant publication in *Nature Communications* once the authors have considered and satisfactorily addressed the above issues.

Response to Reviewers: NCOMMS-23-18372-T

Christopher Lamb, Yicheng Tang, Robert Davis, and Ananda Roy

Report of Reviewer #1

In the following we address the reviewer's comments/suggestions one by one. We reproduce the reviewer's comments in (blue) italics, with our responses in (black) roman. The explicit changes to the manuscript are highlighted as bullet points.

The paper carries out a digital quantum simulation of basic quantum field theory models mapped onto an Ising spin model with transverse field.

We thank the reviewer for his/her reading of our manuscript.

In my view, the most noteworthy aspect of the paper is the complexity of the experiments carried out, with 12 and 20 qubits, in this context of quantum simulation of QFTs. Naturally, this is based on employing the available IBM resources, and not so much on experimental technology of the coauthors' group.

We agree that one of the most noteworthy aspects of this work is the complexity of the experiments carried out on 12 and 20 qubits. It is exciting to have the possibility of verifying QFT predictions with modern quantum simulators. It serves as a precious first step towards using these quantum simulators for addressing questions which could not be addressed with classical computers.

As the referee notes, performing controlled experiments on 12-20 qubits is incredibly challenging and a decade ago, this would be impossible to perform for anyone but a select few experimental groups across the globe. This work was made possible due to the technological advancements in the field of superconducting quantum circuits which have been implemented on IBM's machines. However, we do not believe that this is a 'shortcoming' of our work. That such experiments can in-fact be done by a larger community of scientists is a testament to the overall belief in the field of quantum information that making quantum technologies more widely available will lead to faster progress.

At the same time, the current approach highlights the state of the field, where systems of 10-100 qubits can be reliably manufactured and it is up to the rest of the scientific community to find interesting questions to answer using these systems. This work precisely performs this task where it demonstrates that current quantum simulators can compute the energy spectrum of a strongly-interacting, one-dimensional QFT.

Finally, we bring we to the referee's attention the updated Figs. 2, 3,4 of the current version of the manuscript. To mitigate the effects of decoherence and also address the other referee's comments, we recently implemented a pulse-efficient quantum circuit scheme (Refs. 1,2 in the supplementary methods) and redid the entire set of experiments. This has led to substantial improvement in the meson spectrum computation, as is visible in Figs. 2,3 of the main text.

- Updated Figs. 2,3,4 of the manuscript with recently obtained data with pulse-scaled quantum circuits and updated the corresponding text.

Moreover, quantum simulating an Ising model is not such a novel achievement. Several experiments have been already carried out, both in trapped ions and superconducting circuits, of Ising-like models

with digital quantum simulators, such as, e.g., experiments in the Innsbruck trapped ion lab, as well as in ETH and Google superconducting circuit labs.

We respectfully disagree with the referee's assessment that the work performed in this work is not a novel achievement.

Indeed, as also mentioned in the manuscript, earlier works showed signatures of confinement for certain set of parameters. However, the goal of this work is not to show signatures of confinement. The goal of this work is to show that *current quantum simulators can in-fact compute the energy spectrum of this strongly-interacting QFT* for a wide range of parameter choices. This is also highlighted in the title of the work which includes the phrase "Ising meson spectroscopy", in the abstract, on pages 2,3 and the conclusion.

We analyzed the Ising case because it is simplest model which exhibits the nontrivial confinement properties. However, the same experiments can in fact be done for a wide range of one-dimensional models. In the Supplementary Note II, we describe the results for the XY model, which is a nontrivial generalization of the Ising case.

At a more fundamental level, as also explained in the manuscript (see pages 2, 4), this work emphasizes that simulation of bosonic QFTs on current quantum hardware is much better accomplished using spin-chain regularizations, where the bosonic field arises after bosonization. This is, in contrast, to the initial ideas (Refs. 21-24) where the bosonic field is directly discretized using registers of qubits – which would lead to a very large overhead in a practical implementation. Thus, the approach advocated in this work is precious in obtaining predictions for exotic non-perturbative phenomena which would otherwise remain elusive.

According to my previous comments, in my view this paper would be more suitable for an archival journal.:

We respectfully disagree with the referee's assessment of our manuscript. We hope our arguments convince the referee of the scientific importance of this work and its novelties.

Report of Reviewer #2

In the following we address the reviewer's comments/suggestions one by one. We reproduce the reviewer's comments in (blue) italics, with our responses in (black) roman. The explicit changes to the manuscript are highlighted as bullet points.

This work considers the potential of NISQ devices as simulators of non-perturbative quantum field theories. While problems directly relevant to high energy physics, such as QCD, are way beyond the capabilities of present quantum computers, analogous problems highly relevant to condensed matter physics can be studied in 1+1 dimensional systems. The authors choose a prominent example of these systems, namely, the one-dimensional quantum Ising model, known to have confinement analogous to QCD in its ferromagnetic phase. The resulting meson spectrum can be extracted from quench spectroscopy. The quantum simulation is tested and benchmarked against the theoretical results using the Qiskit package, then implemented on the NISQ hardware.

We thank the reviewer for his/her reading of our manuscript.

The problem addressed by the authors is timely and important; the results are interesting and substantially advance our understanding. The paper is well-written, the exposition is clear, and the authors used state-of-the-art methods to investigate the problem.

We thank the reviewer for his/her positive evaluation of our manuscript.

In my opinion, before eventual acceptance, the authors should consider the following issues and implement the necessary changes to address them:

1. The authors claim that in the NISQ system, the mass of the first meson can always be unambiguously obtained. However, Fig. 2 in itself does not clearly support this statement. Supposing that one only has the experimental data, it needs to be demonstrated that standard peak finding and fitting algorithms do indeed locate the correct position of the first meson peak and also do the same for other data depicted in Fig. 3. An additional question is whether a peak finder algorithm would also produce false positives.

I recommend that the authors perform such an analysis and add the relevant results, as this would make the support of their main statements much more robust.

Indeed, the referee is correct in pointing out that the quench to $g = 1.0$, $h = 0.3$ (right most panel of Fig. 2) does show that the lowest meson mass is not clearly identifiable. This is, in contrast, to the rest of the analyzed set of parameters.

There are two main reasons for this discrepancy. First, for constant h (case considered in the manuscript), as the parameter g approaches 1, the gap of the system diminishes (see Fig. 3). This leads to increased sensitivity to the trotterization and decoherence errors. Second, for the case $g = 1.0$, $h = 0.3$, the parameters of the lattice model are such that the energy difference between the first and third meson energies (e_{13}) is extremely close to the lowest meson energy (e_1). This makes it particularly hard to identify one from the other. This is already visible for the exact computation (with $\Delta t = 0.01$, green curve in Fig. 2) and noiseless quantum Qiskit simulation (with $\Delta t = 0.4$, dashed orange curve in Fig. 2).

The actual experiments are exposed to debilitating effects of decoherence and gate noise, which leads to further deterioration of the results.

However, to address the referee's concern, we chose to redo every single experiment for *all* parameters shown in the previous version of the manuscript. To mitigate the effects of decoherence and gate noise, we implemented a pulse-efficient quantum circuit (Refs. 1,2 of the updated supplementary material). After switching to the Z-basis Hamiltonian (see updated Eq. 1 of the manuscript), following the aforementioned references, we implemented a pulse-efficient scheme for the R_{ZZ} gate needed for the quench experiment. This led to improved data for the meson spectrum for a wider range of parameters including the case of $g = 1.0$, $h = 0.3$. A direct comparison of the data obtained with and without pulse-efficient implementation is shown now in Fig. 3, which clearly demonstrates the benefits of this scheme (not the absence of orange downward triangles for the first three values of η).

Note that the experiments do not unambiguously identify the lowest meson mass for the case $g = 1.0$, $h = 0.3$, due to the proximity of the numerical values of e_{13} and e_1 . However, this is to be expected since even the exact computation data is unable to resolve this difference.

Changes to the manuscript:

- Updated Hamiltonian in Eq. 1 and corresponding Fig. 1 to reflect the new quench evolution.
- Updated Figs. 2, 3, 4 and corresponding captions with newly obtained data based on pulse-efficient quantum circuits.
- Additional clarification on the robustness of lowest meson energy computation below Fig. 3.
- Details of the pulse-scaling scheme in Supplementary Note I.

2. I also find some crucial omissions in the literature considered in the paper; their inclusion is also relevant to emphasize the scope of the present work and future avenues of exploration.

We apologize for the inadvertent omission of the references pointed out by the referee. We have added them in the current version of the manuscript.

The first application of quench spectroscopy to this model appeared in the paper by Rakovszky et al. in Nuclear Physics B 911 (2016) 805-845, in which both the spin chain problem and the corresponding QFT was treated, using iTEBD for the spin chain and TCSA for the QFT.

Added Ref. 37.

An additional phenomenon of interest is Bloch oscillations and Stark localisation, which was addressed by A. Lerose et al. in Phys. Rev. B 102, 041118(R) (2020), and by O. Pomponio et al. in SciPost Phys. 12, 061 (2022). In particular, the latter work performs quench spectroscopy to reveal the characteristic frequency signature of Bloch oscillations. This shows that this phenomenon can also be studied with the methods of the present work, and it would certainly be very promising to consider it for follow-up investigations.

The authors also mention the false vacuum decay and an interesting physical effect in this system. This was considered on the spin chain in G. Lagnese et al., Phys. Rev. B 104, L201106 (2021) and in the QFT in

M. Lencses et al., Phys. Rev. D 106, 105003 (2022). This leads to another interesting question: whether the NISQ hardware used in this work can address the false vacuum decay.

Indeed, it can. We will report on this in an upcoming work.

Added the sentence on page 4 of the manuscript:

- This can lead to investigation of a wide range of exotic QFT phenomena ranging from false-vacuum de- cays [51, 52] to Bloch oscillations [53, 54] in table-top quantum simulation experiments.

In conclusion, I think the results warrant publication in Nature Communications once the authors have considered and satisfactorily addressed the above issues.

We hope we have addressed the referee's concerns adequately. The newly-implemented pulse-scaled circuits revealed experimental data that is entirely consistent with exact numerical predictions for the same lattice model. The data was obtained over multiple days and averaged over multiple runs and verified on two different quantum simulators. We hope our new results convince the referee of the credibility of the experimental data.

REVIEWER COMMENTS

Reviewer #1 (Remarks to the Author):

I have read the reply by the authors and the modifications to the manuscript, and I do not see any substantial advance since my previous referral that may make this manuscript deserve publication in this journal. The model simulated is still the one-dimensional Ising model, which has been carried out extensively in experiments (by the way, the XY model is even easier to be carried out, as in superconducting circuits is the natural Hamiltonian appearing when one eliminates the mw mode, and in trapped ions it can be done with a red and blue sideband, without any further technical difficulty than the Ising. Both models, Ising and XY, are stoquastic, namely, efficiently simulatable classically. It is the Heisenberg, really, the one that becomes more difficult to analyze both numerically and experimentally), the only novel issue is the fact that the number of qubits is about a dozen, which is not enabled by this group but by IBM technology. About dozen qubits is also straightforwardly numerically simulated even by a laptop, for any spin Hamiltonian, even non-stoquastic ones. Therefore, I do not mean this paper is not good, it is just not the kind of paper that, in my view, should be published in a high-profile journal. Thus, my previous referral still holds the same.

Reviewer #2 (Remarks to the Author):

I am satisfied with the answers provided by the authors to my previous review and recommend the work for publication in Nature Communications.

Reviewer #3 (Remarks to the Author):

Dear editors and authors,

Quantum simulation is a promising approach to unravel non-perturbative dynamics of quantum field theories (QFTs) while resources of quantum computers are currently limited to Noisy Intermediate Scale Quantum (NISQ) devices. Therefore it is important to explore and demonstrate the potential of NISQ devices in various problems including QFTs. In this manuscript, the authors implemented

quenched time evolution of the one-dimensional Ising model on IBM's quantum devices to estimate its energy spectrum with quantum error mitigation. In particular, the authors computed an expectation value of γ -spin averaged over the space under the time evolved state, performed Fourier transformation and estimated the energy spectrum in the zero momentum sector from peaks of the absolute value of the Fourier transformation.

The authors found that the locations of peaks obtained by the NISQ simulation are the correct values despite the expectation value itself has quite different values between the results obtained by the NISQ simulation and noiseless Qiskit simulation. This seems to imply that the locations of the peaks are somehow robust against errors of the NISQ devices at least in the Ising model with the parameter region under consideration. Thinking of applications of this method to more general problems, I am wondering in what situations this feature appears. In other words, I am not sure whether this feature is specific to the Ising model, the NISQ devices or both. This point is very important for judging whether or not this manuscript is presenting a great first step to estimate energy spectrum of QFTs that are difficult to analyze by conventional methods.

For this reason, I think that possibility of publication of the manuscript in Nature Communications strongly depends on whether there is a priori argument for the above robustness feature.

Less relevant but non-negligible points:

1. In Fig.2, how the range and resolution of t for taking the data affect accuracy for identifying the peaks?
2. In Fig.3, how were the error bars estimated?
3. In one spatial dimension, confinement is not a non-perturbative phenomenon since even classical electro dynamics exhibits confinement as the Coulomb law gives a linear potential in one spatial dimension. Therefore it is a bit misleading to use just presence of confinement (in one spatial dimension) to argue non-perturbative aspects of QFTs.
4. In supplementary materials, FIGs.2-4 are not referred to in the text, making their roles unclear.
5. In the caption of FIG.2 in supplementary materials, "8,912 shots" could be a typo of "8,192 shots".

Response to Reviewers: NCOMMS-23-18372A

Christopher Lamb, Yicheng Tang, Robert Davis, and Ananda Roy

Report of Reviewer #1

In the following we address the reviewer's comments/suggestions one by one. We reproduce the reviewer's comments in (blue) italics, with our responses in (black) roman. The explicit changes to the manuscript are highlighted as bullet points.

I have read the reply by the authors and the modifications to the manuscript, and I do not see any substantial advance since my previous referral that may make this manuscript deserve publication in this journal. The model simulated is still the one-dimensional Ising model, which has been carried out extensively in experiments (by the way, the XY model is even easier to be carried out, as in superconducting circuits is the natural Hamiltonian appearing when one eliminates the mw mode, and in trapped ions it can be done with a red and blue sideband, without any further technical difficulty than the Ising. Both models, Ising and XY, are stoquastic, namely, efficiently simulatable classically. It is the Heisenberg, really, the one that becomes more difficult to analyze both numerically and experimentally), the only novel issue is the fact that the number of qubits is about a dozen, which is not enabled by this group but by IBM technology.

We had demonstrated the power of NISQ simulators using the Ising case and had also provided results for the XY model. However, the entire point of the paper, as emphasized several times in the manuscript, is that this method can be applied to a wide range of spin chains that are described by one-dimensional, strongly interacting quantum field theories.

To further emphasize this point, we have performed both numerical simulations and quantum simulation experiments on the XYZ spin chain. We assume this is the model that the referee had in mind when he/she used the word 'Heisenberg'. Even with the modest number of qubits used on the IBM device, we were able to obtain results that are in good agreement with the exact diagonalization computations.

About dozen qubits is also straightforwardly numerically simulated even by a laptop, for any spin Hamiltonian, even non-stoquastic ones. Therefore, I do not mean this paper is not good, it is just not the kind of paper that, in my view, should be published in a high-profile journal. Thus, my previous referral still holds the same..

The point of the paper is to describe the potential of NISQ simulators. As we mention already in the abstract, *Our results demonstrate that digital quantum simulation in the NISQ era has the potential to be a viable alternative to numerical techniques such as density matrix renormalization group or the truncated conformal space methods for analyzing QFTs.*

There is no claim in the current paper about the simulator performing tasks that were beyond the classical computing paradigm.

Currently, experiments are being performed on quantum simulators built with thousands of qubits. We are at the edge of exploring new frontiers in physics and we firmly believe our approach to use quantum spin chains and non-equilibrium dynamics to compute spectra of strongly-interacting quantum field theories will have widespread applicability in the coming years.

Report of Reviewer #3

In the following we address the reviewer's comments/suggestions one by one. We reproduce the reviewer's comments in (blue) italics, with our responses in (black) roman. The explicit changes to the manuscript are highlighted as bullet points.

Dear editors and authors,

Quantum simulation is a promising approach to unravel non-perturbative dynamics of quantum field theories (QFTs) while resources of quantum computers are currently limited to Noisy Intermediate Scale Quantum (NISQ) devices. Therefore it is important to explore and demonstrate the potential of NISQ devices in various problems including QFTs. In this manuscript, the authors implemented quenched time evolution of the one-dimensional Ising model on IBM's quantum devices to estimate its energy spectrum with quantum error mitigation. In particular, the authors computed an expectation value of y -spin averaged over the space under the time evolved state, performed Fourier transformation and estimated the energy spectrum in the zero momentum sector from peaks of the absolute value of the Fourier transformation.

We thank the referee for his/her summary of our manuscript.

The authors found that the locations of peaks obtained by the NISQ simulation are the correct values despite the expectation value itself has quite different values between the results obtained by the NISQ simulation and noiseless Qiskit simulation. This seems to imply that the locations of the peaks are somehow robust against errors of the NISQ devices at least in the Ising model with the parameter region under consideration. Thinking of applications of this method to more general problems, I am wondering in what situations this feature appears. In other words, I am not sure whether this feature is specific to the Ising model, the NISQ devices or both. This point is very important for judging whether or not this manuscript is presenting a great first step to estimate energy spectrum of QFTs that are difficult to analyze by conventional methods.

Indeed, the referee raises a very good point. The locations of the peaks are determined by the frequencies in the obtained time-series data. Therefore, even though the amplitude of the time-series data is severely damped due to the gate errors and decoherence, the frequency information is still accessible.

However, this property is not just a lucky characteristic of the model considered. To demonstrate this fact, we performed quantum simulation experiments for Baxter's XYZ chain in a longitudinal field, whose continuum limit is described by a perturbed sine-Gordon model. This model is not free-fermionic and only recently have the mesonic excitation been quantitatively analyzed in a different setup (A. Roy and S. Lukyanov, Nature Communications volume 14, Article number: 7433 (2023)).

Here, we showed that IBM's NISQ simulators using 20 qubits are able to capture the lowest mesons of this perturbed model. This required designing a quantum simulation protocol using varying levels of optimization, all of which is described in detail in the revised supplementary material, Sec. II. We will

report additional material on related field theory models in a more expository manuscript in the coming months.

- Added on page 3: We note that even though the obtained amplitude of the time-series data decays rapidly due to decoherence and gate errors, the frequency, which contains the information of the meson energies, is still discernible. This is true also for other analyzed models, see Supplementary Note II.
- Changed on page 4, second paragraph on the right: ... For details of the simulation protocol and corresponding results, see Supplementary Note II....
- Added new results and methodology discussion in Supplementary Note II.

For this reason, I think that possibility of publication of the manuscript in Nature Communications strongly depends on whether there is a priori argument for the above robustness feature.

We hope we have addressed the referee's concern. We do believe that our approach, which combines spin chain regularization with non-equilibrium dynamics, to extract spectral information of quantum field theories is general and will have wide-spread applicability in the coming years. In fact, the conclusion of this paper describes how NISQ simulators with larger number of qubits will lead to exploration of new frontiers in physics. Since the writing of this paragraph, IBM has launched quantum simulators with 1121 qubits and even larger chips are in the works – we believe that our approach will lead to exploration of a wide range of non-perturbative phenomena that have so far eluded quantitative investigation.

Less relevant but non-negligible points:

1. In Fig.2, how the range and resolution of t for taking the data affect accuracy for identifying the peaks?

The range of t is chosen to be as large as possible, but as seen from Fig. 2, much of the data after $t = 15$ does not contain usable information. The resolution is chosen by performing noiseless simulations using Qiskit and checking if the agreement is reasonable with much smaller trotter steps. As shown by the excellent agreement of the green solid and orange dashed curves, the errors due to the trotterization are small compared to the errors due to decoherence. This is also explained on page 3, left column, paragraph 3.

2. In Fig.3, how were the error bars estimated?

The error bars are estimated based on the spectra resolution of $2\pi/t$, where t = time during the data was acquired.

- Added relevant information in the captions of Supplementary Figures 3 and 6.

3. In one spatial dimension, confinement is not a non-perturbative phenomenon since even classical electro dynamics exhibits confinement as the Coulomb law gives a linear potential in one spatial dimension. Therefore it is a bit misleading to use just presence of confinement (in one spatial dimension) to argue non-perturbative aspects of QFTs.

We do probe non-perturbative spectral properties of the corresponding quantum field theories. The similarities of the Ising model (and the related XYZ model) with t'Hooft's model of quantum chromodynamics have been noticed in earlier works of Fonseca and Zamolodchikov (Ref. 33 and Ref. 34).

We agree with the referee that a more compelling case for the power of quantum simulators will be made by the analysis of two-dimensional models. At the time of writing of the manuscript, no such quantum simulators that could tackle this task. We are currently working on embedding 2D quantum models in quantum simulators and we will report on our finding in the near future.

4. In supplementary materials, FIGs.2-4 are not referred to in the text, making their roles unclear.

We have remedied this in the current version of the manuscript.

5. In the caption of FIG.2 in supplementary materials, "8,912 shots" could be a typo of "8,192 shots".

We have corrected the typo in the current version of the manuscript.

We hope we have addressed the referee's concern adequately. We thank the referee for his/her insightful comments. We trust our paper is now suitable for Nature Communications.

REVIEWERS' COMMENTS

Reviewer #1 (Remarks to the Author):

I still have similar opinion than in my previous referrals of this manuscript, but as the other referees are more positive about publication, and the authors have related their calculations to extensions obeying the XYZ/Heisenberg model, which is less trivial/more interesting, I do not want to block publication. If the other two referees recommend publication, just go ahead.

Reviewer #3 (Remarks to the Author):

I think that the authors properly adressed my concerns. I would recommend the manuscript for publication in Nature Communications.